# Ikaros family zinc finger 1 regulates dendritic cell development and function in humans

Urszula Cytlak [1], Anastasia Resteu[1], Delfien Bogaert[2,3,4,5,6], Hye Sun Kuehn[7], Thomas Altmann[1,8], Andrew Gennery[1,8], Graham Jackson[8,9], Attila Kumanovics[10], Karl V. Voelkerding[10], Seraina Prader[11], Melissa Dullaers [2,5,12], Janine Reichenbach[11,13,14,15], Harry Hill[10,16], Filomeen Haerynck[2,3,5], Sergio D. Rosenzweig[7], Matthew Collin[1,8] & Venetia Bigley [1,8]

Ikaros family zinc finger 1 (IKZF1) is a haematopoietic transcription factor required for mammalian B-cell development. IKZF1 deficiency also reduces plasmacytoid dendritic cell (pDC) numbers in mice, but its effects on human DC development are unknown. Here we show that heterozygous mutation of *IKZF1* in human decreases pDC numbers and expands conventional DC1 (cDC1). Lenalidomide, a drug that induces proteosomal degradation of IKZF1, also decreases pDC numbers in vivo, and reduces the ratio of pDC/cDC1 differentiated from progenitor cells in vitro in a dose-dependent manner. In addition, non-classical monocytes are reduced by IKZF1 deficiency in vivo. DC and monocytes from patients with IKZF1 deficiency or lenalidomide-treated cultures secrete less IFN-α, TNF and IL-12. These results indicate that human DC development and function are regulated by IKZF1, providing further insights into the consequences of *IKZF1* mutation on immune function and the mechanism of immunomodulation by lenalidomide.

[1] Institute of Cellular Medicine, Newcastle University, Framlington Place, Newcastle upon Tyne NE2 4HH, UK. [2] Primary Immunodeficiency Research Lab, Department of Pulmonary Medicine, Ghent University Hospital, C. Heymanslaan 10, 9000 Ghent Belgium. [3] Department of Paediatrics Division of Paediatric Immunology and Pulmonology, Ghent University Hospital, C. Heymanslaan 10, 9000 Ghent Belgium. [4] Center for Medical Genetics, Ghent University Hospital, C. Heymanslaan 10, 9000 Ghent Belgium. [5] Center for Primary Immunodeficiency Jeffrey Modell Diagnosis and Research Centre, Ghent University Hospital, Heymanslaan 10, 9000 Ghent Belgium. [6] Laboratory of Immunoregulation, VIB Inflammation Research Center, Technologiepark 927, 9052 Ghent Belgium. [7] Department of Laboratory Medicine NIH Clinical Center, National Institutes of Health, 10 Center Drive−MSC 1508, Bethesda, MD 20892-1508, USA. [8] Newcastle upon Tyne Hospitals NHS Foundation Trust, Freeman Road, Newcastle upon Tyne NE7 7DN, UK. [9] Northern Institute for Cancer Research, Newcastle University, Paul O'Gorman Building, Framlington Place, Newcastle upon Tyne NE2 4AD, UK. [10] Department of Pathology, University of Utah, 50 North Medical Drive, Salt Lake City, UT 84132, USA. [11] Division of Immunology, University Children's Hospital Zurich, Steinwiesstrasse 75, 8032 Zurich Switzerland. [12] Department of Internal Medicine, Ghent University, De Pintelaan 185, 9000 Ghent Belgium. [13] Children's Research Centre, University Children's Hospital Zurich, Steinwiesstrasse 75, 8032 Zurich Switzerland. [14] Institute for Regenerative Medicine (IREM) Associated Group, University of Zurich, Wagistrasse 12, 8952 Schlieren, Zurich, Switzerland. [15] Competence Center for Applied Biotechnology and Molecular Medicine (CABMM), University of Zurich, Irchel Campus, Winterthurerstr. 190, 8057 Zurich Switzerland. [16] Department of Paediatrics and Medicine, University of Utah, 50 North Medical Drive, Salt Lake City, UT 84132, USA. These authors contributed equally: Matthew Collin, Venetia Bigley. Correspondence and requests for materials should be addressed to V.B. (email: venetia.bigley@ncl.ac.uk)

Effective immunity requires the coordinated development and response of immune cells. This process is orchestrated by transcription factors (TFs), which may act in multiple lineages and govern the expression of both differentiation and functional gene sets. The in vivo functions of specific TFs may be interrogated through the study of primary immunodeficiencies resulting from germline mutations, an approach which offers a wealth of biological insights[1,2].

Dendritic cells (DCs) initiate tolerance or immunity through presentation of antigen and stimulation of naive T cells[3]. In addition, they regulate a range of leukocyte responses including B-cell survival[4] and class switching[5], natural killer cell proliferation and homeostasis[6] and monocyte and neutrophil chemotaxis[7]. DCs consist of two main subsets, plasmacytoid DCs (pDCs) and myeloid or conventional DCs (cDCs), each associated with specific functions[8]. Human pDCs express CD123/IL-3R, CD303/BDCA-2 and CD304/BDCA-4 and, in common with pDCs of all species, secrete large amounts of interferon-α (IFN-α) in response to viruses and other pathogens[9]. Two subsets of cDCs are described; cDC1 and cDC2. In humans these are differentiated by the expression of CD141 and CLEC9A (cDC1) or CD1c (cDC2). cDC1 are specialised in antigen cross-presentation to CD8$^+$ T cells, T helper type 1 polarisation of CD4$^+$ T cells and type III IFN production[10]. Human cDC2s are the predominant interleukin-12 (IL-12) secretors, showing plasticity in T-cell polarisation depending on the environmental stimuli[11].

pDCs and cDCs develop independently of monocytes under the control of specific TFs, largely mapped through the analysis of knockout mice[12]. PU.1 and GATA2 are required for specification of all DCs[13], pDCs are dependent upon IRF8 and E2.2[14], cDC1 on IRF8, Id2 and BATF3[15–17] and cDC2 on IRF4[18]. Classical monocytes, expressing CD14 in human (Ly6C in mouse), require KLF4 at the progenitor stage[19]. Non-classical monocytes express CD16 and can arise from conversion of CD14$^+$ monocytes in the periphery[20].

Ikaros family zinc finger 1 (IKZF1) is a zinc finger TF and member of the IKAROS gene family, with prominent roles in lymphocyte development and proliferative responses[21]. Mutation of Ikzf1 has also been shown to have a dose-dependent effect upon DC development in the mouse. Homozygous Ikzf1$^{L/L}$ mice, expressing low levels of wild-type Ikzf1, have a specific defect of pDCs and loss of IFN-α production[22]. The null allele (Ikzf1$^{C/C}$) prevents formation of pDCs and cDC2s, maintaining a reduced population of cDC1s, whereas the dominant negative DNA binding domain mutant (Ikzf1$^{DN/DN}$) lacks all DCs[23]. Together, these results indicate that murine pDCs are most sensitive to Ikzf1 deficiency and cDC1s the least.

The importance of IKZF1 in human biology is illustrated by its pathogenic involvement in autoimmune disease (systemic lupus erythematosus) and haematopoietic malignancies (B-cell acute lymphoblastic leukaemia), including blastic plasmacytoid dendritic cell neoplasms (BPDCNs), characterised by the expression of pDC markers and CD56 on malignant cells[24].

Human germline heterozygous IKZF1 mutations, resulting in haploinsufficiency, cause a variably penetrant combined immunodeficiency associated with progressive attrition of B cells, hypogammaglobulinaemia and skewing of T-cell subsets[25–27]. Clinical manifestations include recurrent or severe respiratory tract infections, autoimmune phenomena and a predisposition to childhood B-cell acute lymphoblastic leukaemia.

IKZF1 is also known to be a key target of thalidomide and its derivatives, used to treat myeloma and 5q-myelodysplasia. It has recently been shown that their therapeutic actions include activation of Cereblon-dependent ubiquitination and proteasomal degradation of IKZF1 and IKZF3[28,29]. Thus, exposure to lenalidomide induces IKZF1 deficiency offering a further opportunity

to manipulate IKZF1 levels in vivo or during differentiation and functional analysis of human cells in vitro.

Prompted by the knowledge that murine pDC development is dependent upon Ikzf1, here we investigate whether IKZF1 mutation or inhibition with lenalidomide causes pDC deficiency in humans, using phenotypic and functional analyses performed on patients with IKZF1 haploinsufficiency, those receiving lenalidomide, or on progenitor cell cultures exposed to lenalidomide in vitro. In addition to pDC deficiency, we observe a relative increase in cDC1 in vivo and in vitro and a loss of non-classical monocytes in vivo. In the presence of IKZF1 deficiency, pDCs produce less IFN-α, pDCs and monocytes secrete less tumor necrosis factor (TNF), and cDC1, although increased, produce less IL-12. These results extend the known functions of IKZF1 to include the regulation of human DC haematopoiesis.

## Results

**IKZF1 haploinsufficiency cohort.** The clinical features, mutations and B-cell phenotype of 20 individuals from 4 families with heterozygous IKZF1 mutations have been previously reported. Families B, C and F were studied by Kuehn et al.[25]. Members of family G have been recently described[27]. Replicate B-cell counts performed on blood taken for this study were congruent with the analyses previously reported. These and further details are summarised in Supplementary Table 1.

**pDC deficiency and cDC1 expansion in IKZF1 mutation.** In order to map the global perturbation of DC haematopoiesis induced by IKZF1 mutation, an unsupervised phenotypic analysis and enumeration was initially conducted using a member of family B (B5) and a control. The FlowSOM algorithm was used to cluster, visualise and compare equal numbers of data from the Lineage$^-$HLA-DR$^+$ (Lin$^-$DR$^+$) CD14$^-$ gate of a 16-colour flow cytometry panel (Supplementary Figure 1a, Supplementary Table 2). This algorithm clusters cells of the same phenotype into populations, each represented as a node. Nodes are presented as coloured metaclusters on a minimal spanning tree, defining the relationships between nodes[30] (Fig. 1a). With an equal number of events from the affected and unaffected individual, FlowSOM identified 6 metaclusters: 1 corresponding to CD123$^+$ pDCs (blue nodes), 1 corresponding to CD141$^+$BTLA$^+$ cDC1s (turquoise nodes) and the remaining 4 corresponding to CD11c$^+$CD1c$^+$CD2$^+$ cDC2s (red nodes). Within metaclusters it was possible to discern a CD2$^+$ node of pDCs, and variable expression of CD2, CD5 and CD1c forming discrete nodes of cDC2s. To analyse the differences between the affected and unaffected individuals, the relative contribution of cells from each individual to each node was assessed (Fig. 1b). IKZF1 mutation was associated with an overall relative reduction in pDCs, but increase in cDC1s. There was an altered distribution of cells within the cDC2 nodes. The relative loss of pDCs was consistent with the highest expression of IKZF1 mRNA and protein in this subset of DCs (Fig. 1c, d, Supplementary Figure 1b). For comparison, IKZF3 expression is shown (Fig. 1c).

**DC subset skewing in IKZF1 haploinsufficiency.** The relative decrease in pDCs and expansion of myeloid cDC1s and cDC2s was further defined by performing absolute whole blood counts on all patients compared with healthy controls. Within the CD3$^-$ mononuclear cell gate, the HLA-DR$^+$CD4$^+$ population contained CD14$^+$ classical and CD16$^+$ non$^-$ classical monocytes, CD123$^+$ pDC, CD141$^+$ cDC1 and CD1c$^+$ cDC2 (Fig. 2a, Supplementary Figure 1c). All affected individuals had a profound reduction in pDCs but expansion in cDC1s ($p < 0.05$, by two-tailed Mann–Whitney U-test) (Fig. 2b). In absolute counts, cDC2s

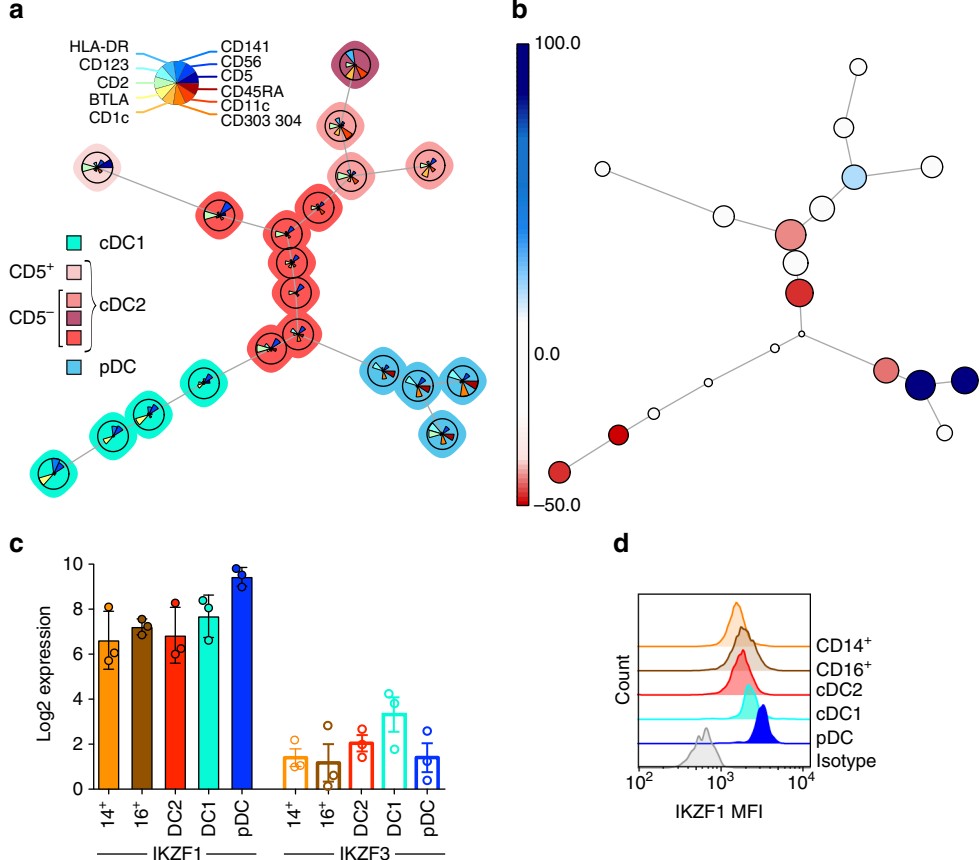

**Fig. 1** FlowSOM analysis of PB reveals a reduced proportion of pDC but increased cDC1 in a patient carrying *IKZF1* mutation. **a** Minimal spanning tree visualisation of a self-organising map using compensated flow cytometric data from a family B member (B5) compared to a travel control (equal number of events). Data were taken from the lineage (CD3, CD7, CD14, CD16, CD19, CD20)-negative HLA-DR-positive gate. FlowSOM nodes represent clusters of cells. Metaclusters of the nodes, determined by the map, are represented by the background colour of the nodes. Star charts represent the mean marker value of cells in that node with the height of each part corresponding to marker intensity. **b** Comparison between samples used to generate the map; size of nodes represents proportional number of cells in each node, colour represents proportional differences between samples with red and blue indicating higher or lower numbers in *IKZF1* mutation compared to wild type, respectively. **c** mRNA and **d** protein expression of IKZF1 in healthy donor monocytes and dendritic cells by NanoString gene expression analysis of FACS sorted cells or intracellular flow cytometry, respectively ($n = 3$ donors for each experiment). *IKZF3* mRNA expression is shown for comparison in **c**. cDC1/2, conventional dendritic cell 1/2; pDC, plasmacytoid dendritic cell; 14+ CD14+, classical monocyte; 16+ CD16+, non-classical monocyte

were not affected. Classical monocytes were also in the normal range but non-classical monocytes were reduced, even in patients who had received no therapy ($p < 0.05$, by two-tailed Mann–Whitney *U*-test). Quantitative changes were less pronounced in family F carrying a multi-gene deletion on chromosome 7, encompassing *IKZF1*.

There was no effect of age on the DC phenotype, which was present in clinically asymptomatic and symptomatic individuals (Fig. 2c). An increased proportion of CD56+ cells was confirmed in all three DC subsets (Fig. 2d).

**IKZF1 deficiency and pDC depletion in lenalidomide treatment.** An independent verification of the effect of *IKZF1* mutation on pDC development was sought through the analysis of patients receiving lenalidomide for haematological malignancy. Patient characteristics are summarised in Supplementary Table 3. Owing to the fact that a range of lenalidomide dosing schedules are employed, the level of IKZF1 protein was first quantified by intracellular flow cytometry of peripheral blood B cells. Lenalidomide treatment on the day of sample analysis resulted in a reduction in B-cell IKZF1 protein, comparable to that seen in heterozygous *IKZF1* mutation in family G (Fig. 3a). Patients on

maintenance lenalidomide sampled between treatment courses had intermediate levels that correlated with dose (Fig. 3b). A negative correlation between lenalidomide dose and the number of circulating B cells was also observed. (Fig. 3c).

Absolute pDC counts showed a significant positive correlation with IKZF1 protein level in 24 patients treated with lenalidomide ($n = 22$) or pomalidomide ($n = 2$) by linear regression analysis ($r^2 = 0.6561$, $p < 0.0001$). The inclusion of 4 healthy controls and 3 affected family G members did not significantly alter the slope or significance of the linear regression analysis ($r^2 = 0.6541$, $p < 0.0001$) (Fig. 3d).

Patients on lenalidomide also showed a reduction in CD16+ non-classical monocytes, reaching statistical significance ($p = 0.02$ by Mann–Whitney *U*-test) in those with the lowest IKZF1 protein levels (defined as R2 in Fig. 3d). Unlike the families with *IKZF1* mutation, lenalidomide treatment was associated with a slight depression in cDC2 and no increase in cDC1 (Fig. 3e) compared to healthy controls.

Lenalidomide also causes depletion of IKZF3, but IKZF1 is expressed at more than 100 times the level of IKZF3 in human pDCs (mean log2 9.8 and 1.4 respectively; Fig. 1c).

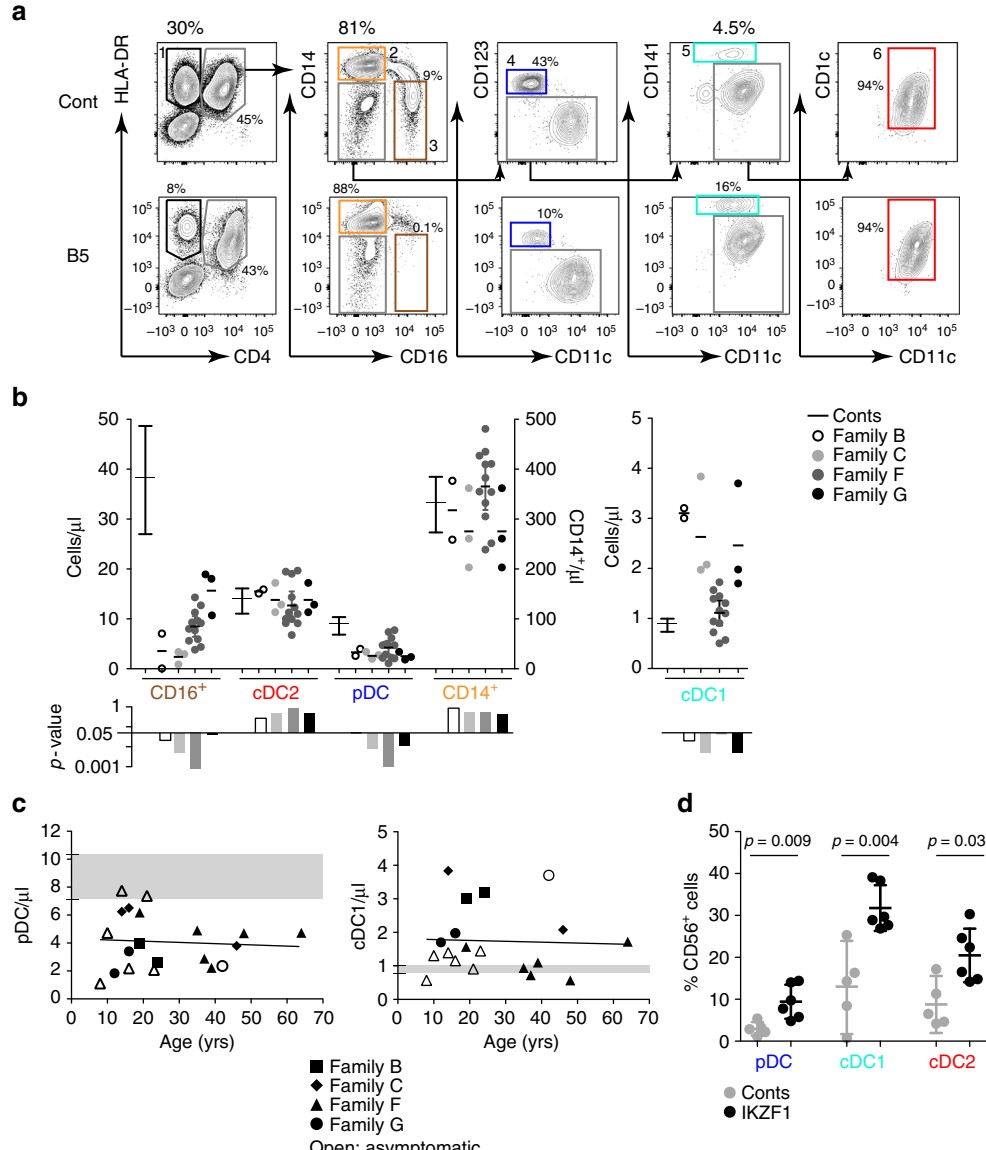

**Fig. 2** Reduced pDC and non-classical monocytes but expanded cDC1 in the IKZF1 haploinsufficiency cohort. **a** Flow cytometric quantification of PB monocyte and dendritic cell (DC) subsets showing HLA-DR$^+$CD4$^-$ B cells, CD14$^+$ classical (orange) and CD16$^+$ non-classical monocytes (brown), CD123$^+$ plasmacytoid DCs (pDC) (blue), CD141$^+$ conventional DC1 (cDC1) (turquoise) and CD1c$^+$ cDC2 (red) in healthy donor (cont, (control)) and representative individual with *IKZF1* mutation. **b** Absolute counts of monocytes and DCs in 20 individuals from 4 affected families compared to $n = 32$ healthy donors. Bars show mean $\pm$ 95% confidence interval (CI). Histograms show subset-specific *p*-values for each family compared to healthy donors by two-tailed Mann–Whitney *U*-test with significance set at $p < 0.05$. **c** Absolute pDC or cDC1 counts plotted against the age of the individual. Grey zones represent the normal range of healthy controls ($n = 32$), black lines represent linear regression analysis with $p = 0.77$ and $p = 0.86$, respectively. **d** Proportion of cells expressing CD56 in healthy donors ($n = 5$) and individuals carrying *IKZF1* mutation ($n = 6$). Bars represent mean $\pm$ 95% CI. The *p*-values derived from two-tailed Mann–Whitney *U*-test

**Reduced IL-12 and IFN-α production in IKZF1 deficiency.**
Functional defects associated with loss of IKZF1 were investigated by examining intracellular cytokine production by specific DC and monocyte subsets in response to a cocktail of Toll-like receptor (TLR) agonists (polyinosinic:polycytidylic acid (poly(I:C)), lipopolysaccharide (LPS), CL075 and CpG). In healthy control peripheral blood mononuclear cells (PBMCs), no differences were observed in cell-specific production of IFN-α, IL-12 or TNF in response to the relevant single TLR agonist compared to the cocktail (Supplementary Figure 2a).

Individuals with *IKZF1* mutation and healthy control PBMCs with or without exposure to lenalidomide were examined. TNF production by all DC and monocyte subsets was reduced in the presence of *IKZF1* mutation or exposure to lenalidomide (Fig. 4a, b, Supplementary Figure 2b, c). IFN-α production by pDC and monocytes was also abolished or greatly reduced, respectively, on a per-cell basis, especially with *IKZF1* mutation. Production of IL-12 by myeloid cDC1s and cDC2s was also reduced in both conditions. IL-10 was quite strongly induced by lenalidomide in monocytes and myeloid cells but was inversely affected by *IKZF1* mutation.

In an attempt to dissect whether the loss of IL-12 production was a secondary effect of the reduction in IFN-α secretion by pDCs, it was observed that production of IFN-α by healthy control pDCs could be abrogated by ligation of CD303 and CD304 with anti-CD303/4 antibodies. Although exogenous

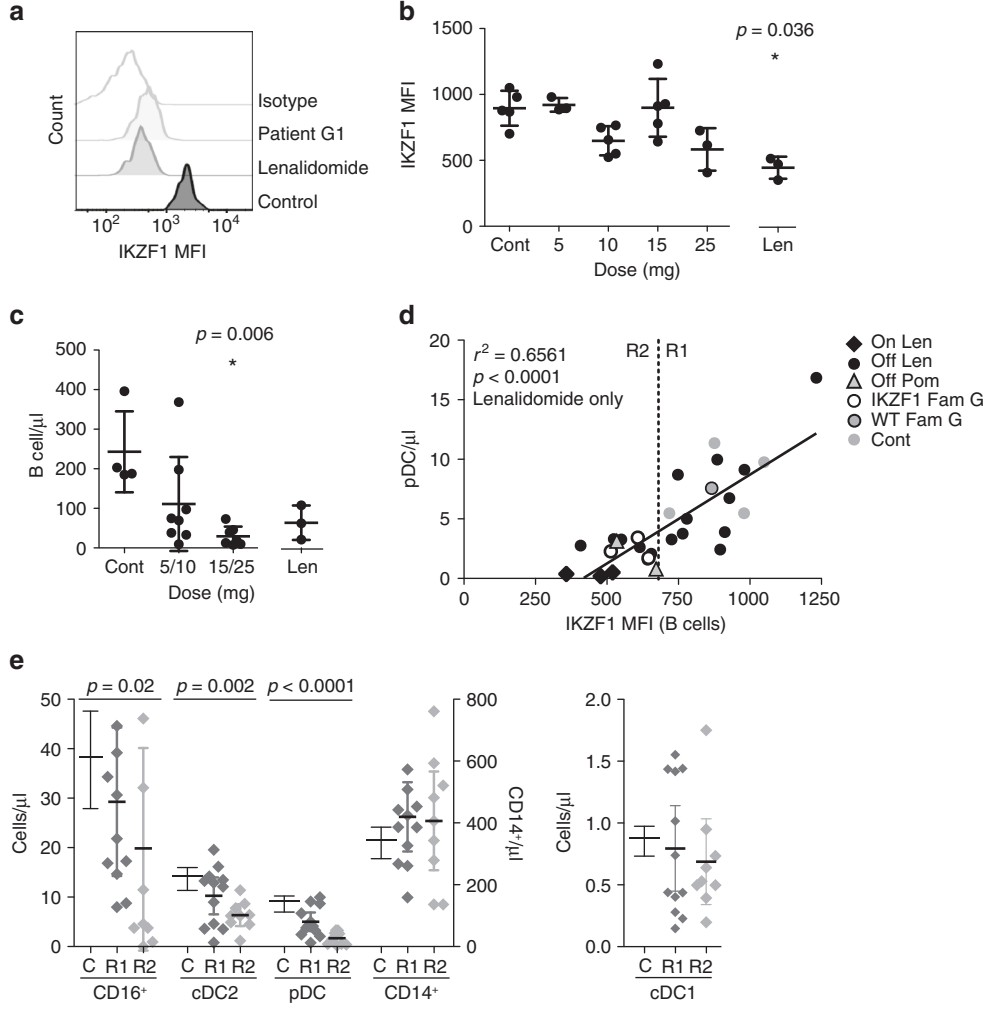

**Fig. 3** Dose-dependent reduction of IKZF1 protein and pDCs in patients receiving lenalidomide treatment. **a** Intracellular flow cytometric quantification (mean fluorescence intensity (MFI)) of IKZF1 protein levels in B cells (identified as SSC[low]Lin[−]DR[+]CD2[−] cells) of a family G member carrying *IKZF1* mutation, patient on lenalidomide treatment, a healthy donor and isotype control. **b** Intracellular IKZF1 level (MFI) in healthy donors ($n = 5$), patients tested at the end of a week off treatment with specified dose of lenalidomide ($n = 16$) (5, 10, 15 or 25 mg daily) and patients taking lenalidomide (Len) on the day of analysis ($n = 3$) (15 mg daily). Bars represent mean ± s.d. *$p = 0.036$ by two-tailed Mann–Whitney $U$-test versus healthy donor controls. **c** Summary of absolute B-cell counts in whole blood from healthy donors ($n = 4$) (Cont), patients tested at the end of a week off treatment with specified dose of lenalidomide ($n = 16$) (5 or 10, 15 or 25 mg daily) and patients taking lenalidomide on the day of analysis ($n = 3$) (15 mg daily). Bars represent mean ± s.d. *$p = 0.006$ Mann–Whitney $U$-test versus healthy donor controls. **d** IKZF1 protein levels (MFI) plotted against absolute plasmacytoid dendritic cell (pDC) counts in affected ($n = 3$) and unaffected ($n = 1$) members of family G (white or grey open circles, respectively), patients after a week off lenalidomide ($n = 22$) (black circles) or taking lenalidomide on the day of analysis ($n = 3$) (black diamond) or healthy donors ($n = 4$) (grey circles). Data from two patients taking pomalidomide (4 mg daily) are displayed (grey triangles). Black line represents linear regression analysis of lenalidomide-treated patients only. Inclusion of family G members and healthy controls in the linear regression analysis resulted in $r^2 = 0.6541$, $p < 0.0001$. Inclusion of pomalidomide-treated patients resulted in $r^2 = 0.6419$, $p < 0.0001$. Dashed line represents IKZF1 MFI = 680 dividing group R1 (IKZF1 MFI > 680) from group R2 (IKZF1 MFI < 680). **e** PB absolute monocyte and DC counts in healthy controls (C) compared to patients from group R1 and R2. The $p$-values from two-tailed Mann–Whitney $U$-test of R2 versus healthy donors

IFN-α had a slightly enhancing effect on the secretion of IL-12 by cDC2, the baseline production of IL-12 by cDC2 was not at all affected when IFN-α production by pDCs was completely blocked, suggesting that the loss of IL-12 production in the preceding experiments was directly attributable to loss of IKZF1 (Fig. 4c).

**IKZF1 deficiency impairs pDC differentiation in vitro.** The effect of IKZF1 deficiency on human DC development was examined in vitro. DC subsets were generated from human bone marrow CD34[+] progenitors after 22 days (D22) of culture in the presence of a lenalidomide titration.

DC subsets were identified by their surface marker expression profile corresponding to blood counterparts: CD11c[+]CD14[+] monocytes, CD141[+]CLEC9A[+] cDC1, CD11c[+]CD1c[+] cDC2 and CD303[+]CD304[+]CD123[+] pDC (Fig. 5a, Supplementary Figure 1d). There was a negative correlation between lenalidomide concentration in the culture and IKZF1 protein level in Lin[−]DR[+] cells at D22 of culture (Fig. 5b). This was associated with a reduction in pDC and cDC2 but increase in cDC1 output. CD14[+] cells were unaffected (Fig. 5c). A reduction in the number of cells generated per input progenitor was seen at lenalidomide concentrations above the published in vivo plasma $C_{max}$ for therapeutic dosing (1.7–2.3 μM)[31] (Fig. 5d), but there remained a clear dose-dependent effect on the cDC1/pDC ratio (Fig. 5e).

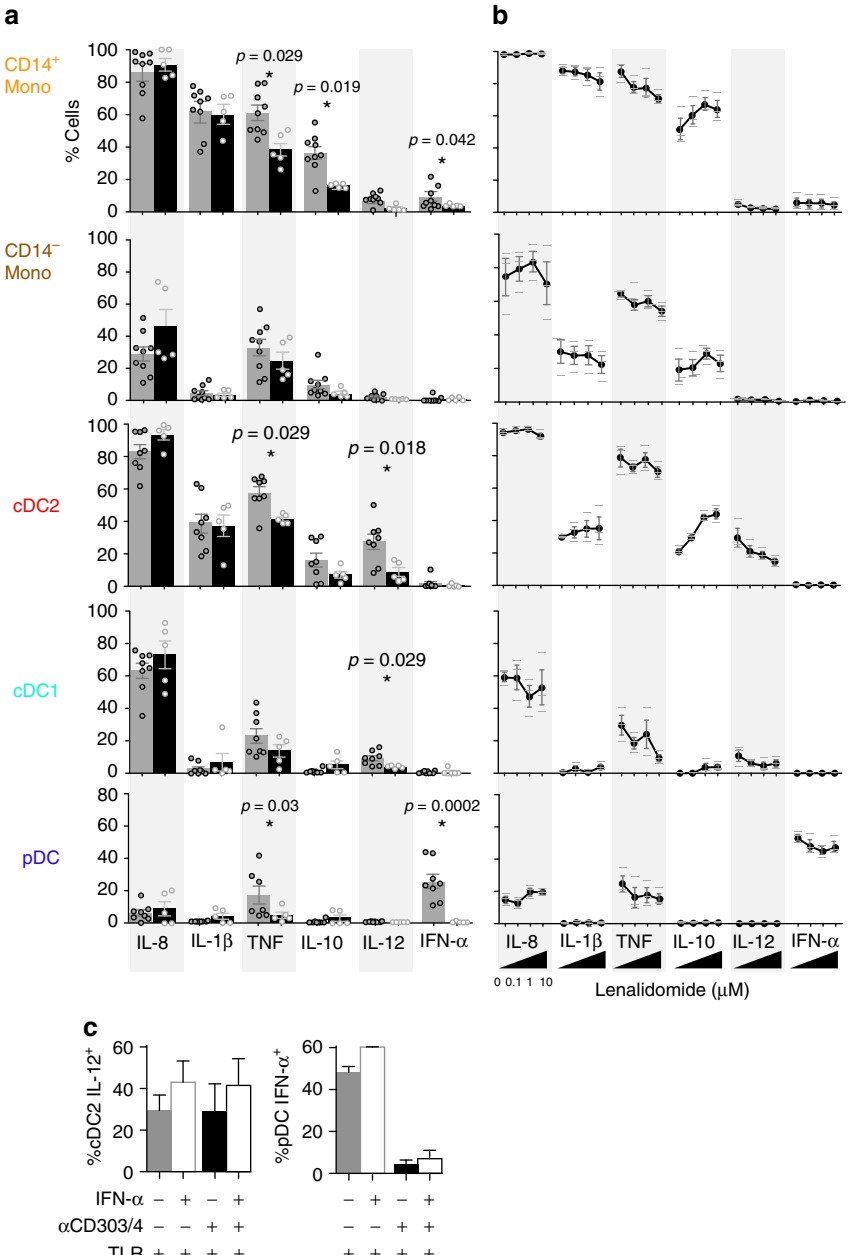

**Fig. 4** Reduced production of TNF, IL-12 and IFN-α in IKZF1 deficiency. **a** Intracellular flow cytometric analysis of cytokine production (% positive cells) in PB CD14+ monocytes (mono), CD14− monocytes and dendritic cell (DC) subsets (cDC2, cDC1, pDC) from $n = 5$ individuals carrying IKZF1 mutation (black bars) or $n = 8$ healthy donors (grey bars) following 14 h of stimulation of PBMCs with a cocktail of TLR agonists (CpG, poly(I:C), CL075, LPS). Histograms represent mean and bars represent s.e.m. Dots represent individuals. The p-values derived from Mann–Whitney U-test of IKZF1 mutation versus healthy donors. **b** Analysis repeated on healthy donor PBMCs in the presence of increasing concentrations of lenalidomide (0, 0.1, 1 or 10 μM). Black dots represent mean, bars represent s.e.m. and lines represent individual data points from $n = 3$ donors in each condition. **c** IL-12 or IFN-α production from cDC2 or pDC, respectively, from healthy donor PBMCs stimulated with TLR cocktail (TLR) and without (solid bars) or with (unfilled bars) the addition of exogenous IFN-α and without (grey) or with (black) anti-CD303 and anti-CD304 antibodies

## Discussion

IKZF1 is a key regulator of haematopoiesis and a critical factor in murine lymphocyte development and function[21]. Normal IKZF1 protein levels are also necessary for the development of IFN-α-producing pDC in mice[22,23]. Recent descriptions of human IKZF1 haploinsufficiency have confirmed its role in human lymphocyte biology but human DC development has not been studied[25–27]. In this study we analysed blood monocytes and DCs from patients ex vivo carrying heterozygous IKZF1 mutations, or treated with lenalidomide, an immunomodulatory drug.

We also probed the effects of IKZF1 deficiency on human DC development and function in vitro.

In keeping with the pleiotropic actions of haematopoietic TFs, IKZF1 deficiency resulted in multi-lineage developmental and functional defects. In addition to the previously described progressive loss of B cells and skewing of T-cell subsets, we found deficiency of pDCs and non-classical monocytes but expansion of cDC1s. Classical monocytes and cDC2 remained numerically unaffected. The near universal finding of this antigen presenting cell phenotype, independent of age, lymphocyte phenotype or

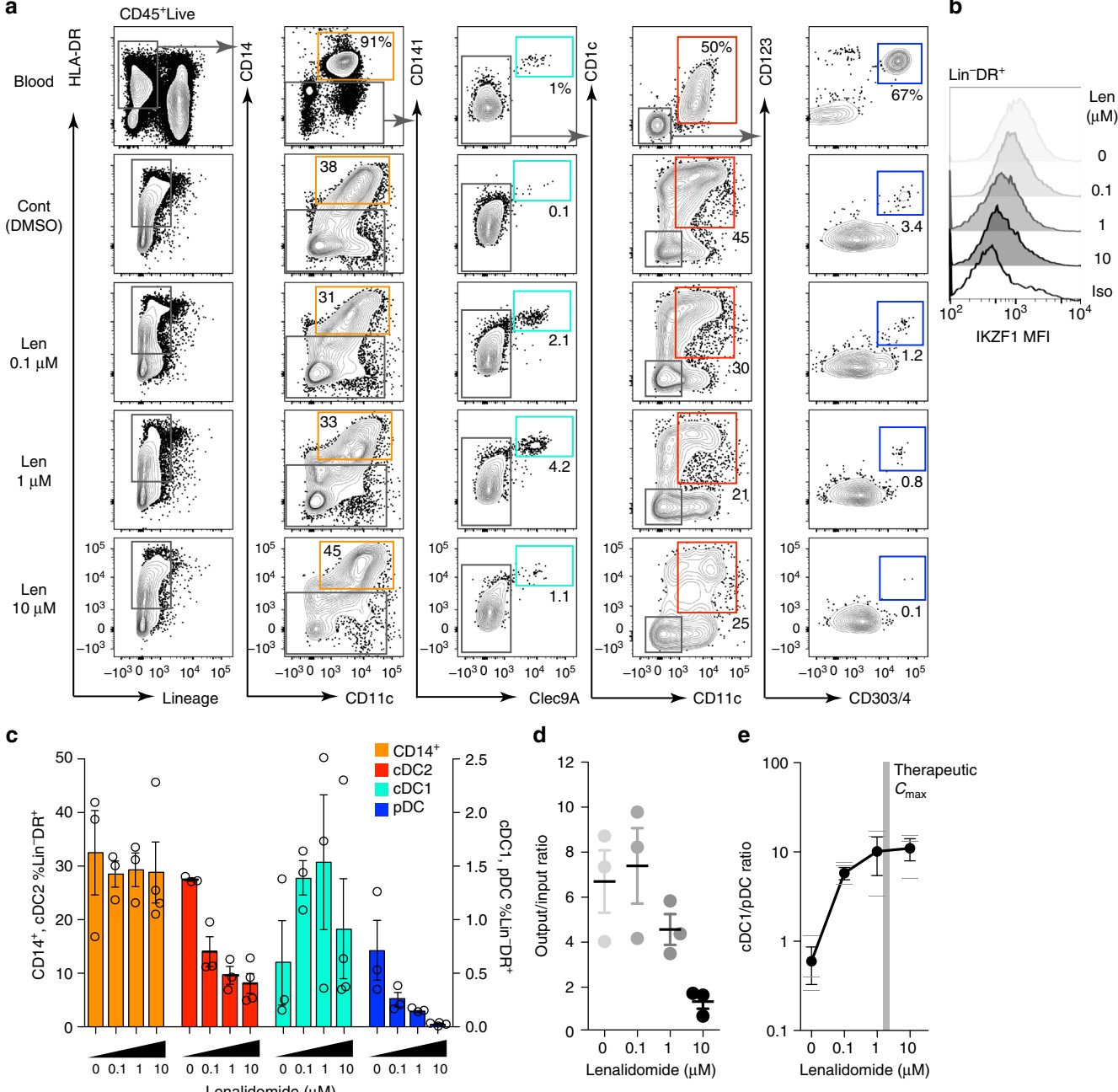

**Fig. 5** Impaired in vitro pDC development in IKZF1 deficiency. **a** Phenotypic flow cytometric analysis of dendritic cell (DC) subsets generated after 22 days of culture in vitro, compared to PB counterparts. Found within the Lineage⁻HLA-DR⁺ gate were CD14⁺CD11c⁺ monocytes, CD141⁺CLEC9A⁺ cDC1s, CD11c⁺CD1c⁺ cDC2s and CD123⁺CD303⁺CD304⁺ pDC. **b** IKZF1 protein levels, determined by flow cytometric analysis and IKZF1 mean fluorescence intensity (MFI), in Lin⁻DR⁺ cells after 22 days of culture in the presence of a lenalidomide titration. **c** Summary of the proportion of in vitro generated Lin⁻DR⁺ cells identified as CD14⁺ monocytes (orange), cDC2 (red), cDC1 (turquoise) and pDC (blue). Bars represent mean and s.e.m. of triplicate wells, dots represent single cultures. **d** Ratio of total number of Lin⁻DR⁺ cells generated in vitro per input CD34⁺ cell in the presence of increasing concentrations of lenalidomide. **e** Lenalidomide dose response of the cDC1 to pDC ratio of in vitro generated cells. Grey zone indicates published human in vivo plasma $C_{max}$ of lenalidomide (450–600 ng/ml or 1.7–2.3 µM). Dots represent mean, bars represent s.e.m. and lines represent individual cultures

clinical status, provides a cellular signature of human *IKZF1* mutation. The quantitative changes were remarkably similar in all individuals with missense proteins (families B and C), or truncated protein (family G[27]), but less severe in members of family F who carry a heterozygous, 11-gene deletion of chromosome 7. In homodimeric proteins, it has been proposed that a heterozygous missense mutation may result in a more severe phenotype than a null allele due to the lower proportion of WT/WT dimers (25% versus 50%, respectively)[32]. However, in the case of family F, a

compensatory effect due to the loss of additional genes cannot be excluded.

The requirement for IKZF1 in human pDC development and function mirrors that seen in the mouse and was supported by its high level of expression in healthy control pDCs. There was no significant increase in absolute number or proportion of cDC1 in mice carrying the heterozygous *Ikzf1* L allele, tested in cohorts of 3 animals[22], representing either a species or mutation-specific difference. Targets of IKZF1, identified by chromatin

immunoprecipitation sequencing[33], include ID2, suppression of which is necessary for pDC development and BATF3, required for cDC1 terminal differentiation. De-repression of these loci due to IKZF1 deficiency is consistent with the observed phenotype of absent pDCs but preserved or expanded cDC1s.

The reduction in non-classical monocytes, to our knowledge, has not been reported in Ikzf1-deficient mice. This finding was independent of therapeutic interventions including intravenous immunoglobulin and corticosteroid treatment, previously reported to result in transient depletion of CD16+ monocytes[34]. Ly6C^low murine monocytes, corresponding to human CD16+ non-classical monocytes, convert from classical monocytes under the control of NOTCH2 signalling stimulated by endothelial cell notch-ligand delta-like 1 (DLL1)[35]. The role of notch signalling in the generation of CD16+ classical monocytes is untested, but it is known that the regulation of notch target genes is IKZF1 dependent in human T cells[36].

The cell-intrinsic effect of IKZF1 mutation on DC phenotype was confirmed in patients receiving therapeutic lenalidomide, known to target IKZF1 for proteosomal degradation[28,29].

Varying lenalidomide dose schedules resulted in a range of IKZF1 levels in vivo, revealing a linear relationship between IKZF1 protein and the frequency of pDCs. Such an in vivo dose-response effect would be difficult to demonstrate from the series of germline mutations that confer idiosyncratic, allele-specific effects upon protein structure and function. Parallel observations on the in vitro generation of DCs from primary bone marrow progenitors showed a lenalidomide dose-dependent decrease in the production of pDCs and increase in cDC1s. Although the increased ratio of cDC1 to pDCs was strikingly similar in the ex vivo analysis of patients with germline IKZF1 mutation and those treated with lenalidomide, cDC1s were not expanded and cDC2s were reduced by the drug. This may be due to the known myelosuppressive effect of lenalidomide as concentrations above the therapeutic $C_{max}$ of lenalidomide resulted in a reduction in the cellular output per input progenitor cell in vitro. In addition, cereblon-dependent suppression of IRF4 by lenalidomide[37] may contribute to the dose-dependent reduction in cDC2 seen in vivo and in vitro. While our data are unable to exclude an effect of IKZF3 deficiency on the DC phenotype in lenalidomide treatment, it is expressed at a much lower level than IKZF1 in human DCs and a role for this factor in DC differentiation has not been described in murine models.

In functional terms, IKZF1 haploinsufficiency resulted in perturbed cell-specific cytokine secretory responses to TLR agonists. Remaining pDCs were unable to secrete IFN-α, production of IL-12 by cDCs was reduced and all cells failed to elaborate as much TNF. A similar pattern was seen in healthy donor DCs exposed to lenalidomide. The reduction in IL-12 secretion contrasted with reports showing that lenalidomide does not compromise IL-12 production from monocyte-derived DCs (moDCs) stimulated with CD40L[38,39]. However, moDCs are not dependent on IKZF1 for development[40] and in vitro stimulation with CD40L triggers IL-12 production through the non-canonical, nuclear factor (NF)-κB (p52/p100) pathway.

Our data are consistent with a direct effect of IKZF1 deficiency upon canonical NF-κB (Rel-A/p50) signalling in which IKZF1 is necessary for the upregulation of Rel-A[41] and is itself upregulated by LPS-TLR4 stimulation[42]. We considered the additional scenario that down regulation of IL-12 might have been an indirect effect of loss of type I IFN production by pDCs, as exogenous IFN augmented IL-12 production[43]. However, CD303/CD304 ligation, which also abrogates IFN-α, failed to reduce IL-12 production and lenalidomide resulted in a similar reduction in IL-12, despite only a modest fall in IFN-α. From these observations we conclude

that lower IL-12 production by cDCs was most likely intrinsic to loss of IKZF1.

The multi-lineage and multi-level influence exerted by haematopoietic TFs complicates the attribution of immunodeficiency resulting from TF mutation to defects in specific immune cell types. In DC deficiency states, the functional diversity of DCs, their combined roles in innate and adaptive immunity and their potential to both activate and tolerise add further complexity. In summarising the consequences of IKZF1 deficiency, pDC dysfunction is likely to play a role. An increased risk of bacterial infection, particularly respiratory infection in the context of germline haploinsufficiency, is consistent with the role of pDC in prompt bacterial clearance and limitation of inflammation in the lung[44], in addition to their known anti-viral properties. Humoral immune responses are also dependent upon pDC function through the promotion of naive and memory B-cell proliferation, plasma cell differentiation and immunoglobulin secretion[45]. This is in keeping with a contribution of pDC deficiency to progressive hypogammaglobulinaemia seen in IKZF1 haploinsufficiency, despite the persistence of plasma cells in tissues[25]. In other settings, pDCs promote peripheral and central tolerance, through induction of natural and induced regulatory T cells and direct suppression of T-cell responses[46]. Related to their tolerogenic role, pDCs in the bone marrow microenvironment have been shown to support multiple myeloma cell growth and mediate myeloma-associated immunodeficiency[47]. The loss of pDCs may therefore promote the development of autoimmunity in IKZF1 haploinsufficiency and confer therapeutic benefit in the treatment of multiple myeloma. These effects are potentially enhanced by an increase in the cDC1/pDC ratio. cDC1s, specialised for cross-presentation of antigen to cytotoxic T cells, are the most potent DCs in immunity to tumours and vaccinations. Consistent with this, in the murine model of multiple myeloma, lenalidomide synergistically enhances the anti-tumour effect of DC vaccines[48] and in myeloma patients, lenalidomide enhances responses to a pneumococcal vaccine[49].

Finally, IKZF1 and pDC are connected in a number of other conditions. IKZF1 is a susceptibility locus in systemic lupus erythematosus, notable for a type I IFN signature and dysregulated pDC function (reviewed in ref. [50]). In BPDCN, frequently involving deletion or loss of function mutations of IKZF1, increased CD56 expression is a hallmark of the neoplastic pDC phenotype[24]. In the studies described here, increased CD56 expression is seen to arise directly from IKZF1 deficiency.

In summary, our data demonstrate that in addition to its critical role in B-cell differentiation, IKZF1 is required for human pDC development and function. Together with the parallel expansion of cDC1s and reduction of non-classical monocytes, this comprehensively defines the cellular signature of IKZF1 haploinsufficiency. DC dysregulation is highly likely to have pathological consequences for immunity in germline IKZF1 mutation but confer additional therapeutic benefit in lenalidomide treatment of plasma cell dyscrasias. In common with other haematopoietic TFs, germline deficiency reveals multi-level and multi-lineage roles in immune cell development and function with effects in B-cell, T-cell, DC and monocyte lineages.

## Methods

**Study approval**. The study was performed in accordance with the Declaration of Helsinki. Written informed consent was obtained from participants, or their parents, prior to recruitment. The study was approved by local review boards: NRES Committee North East–Newcastle and North Tyneside 1, 08/H0906/72; KEK-ZH Nr. 2015-0135; IRB 00029386; Ethical Committee of Ghent University Hospital, 2012/593.

**Patients**. Individuals carrying an IKZF1 mutation, and family members, were recruited at their local medical centres in accordance with local ethical permissions.

All participating patients were included in the study. The family nomenclature (families B, C and F) corresponds to the nomenclature published in ref. [25], with the exception of a newly described family 'G'[27,49,50].

Patients receiving lenalidomide treatment were recruited from a local ambulatory myeloma clinic. There were no specific inclusion or exclusion criteria and analyses from all tested patients were included.

**Flow cytometry and cell sorting**. PBMCs, separated by density centrifugation, were stained in aliquots of $1-3 \times 10^6$ cells in 50 µl of Dulbecco's phosphate-buffered saline with 2% fetal calf serum and 0.4% EDTA. Dead cells, usually <5%, were excluded by 4',6-diamidino-2-phenylindole (Partec) or Zombie (Biolegend). Analysis was performed with an LSRFortessa X-20 and sorting with a FACSAria III (BD Biosciences) running BD FACSDIVA™ 8.0.1 or 8.0 software, respectively. Data were processed with FlowJo 10.1r5 (Tree Star, Inc.). Absolute cell counts were obtained using TruCount™ tubes (BD Biosciences) with 200 µl whole blood and 900 µl of red cell lysis buffer. Intracellular staining was performed after surface staining, lysis and fixation (eBioscience) according to the manufacturer's instructions. Antibodies used are given in Supplementary Table 2.

**FlowSOM analysis**. Flow cytometric analysis of PBMCs from a member of family B (B5) carrying *IKZF1* mutation and a travel control were analysed using an 18-channel (16 fluorochromes, 2 light scatter) panel. Compensated FCS files were manually gated (FlowJo 10.1r5, Treestar, Inc.) to export Lin⁻DR⁺CD14⁻ cells.

FlowSOM 1.7.1 was used for further analysis. From each of the two files, 550 cells were randomly selected. A total of 12 surface markers were used for building the self-organising map (SOM): CD5, CD141, CD123, CD2, BTLA, HLA-DR, CD1c, CD303, CD304, CD11c, CD45RA, and BTLA. CD56 was included in visualisation.

SOM grid dimensions were set to $4 \times 5$ and the resulting SOM visualised in a minimal spanning tree with 20 nodes, corresponding to cell clusters. The maximum number of metaclusters, equivalent to predicted cell types, was set to 15. The software identified 6 metaclusters, represented by background colour of the nodes.

The differences for each node were calculated by subtracting the number of patient cells from the number of healthy control cells.

**DC functional analysis**. The $3 \times 10^6$ PBMCs from healthy donors or individuals carrying heterozygous *IKZF1* mutation were cultured in the presence poly(I:C) (10 µg/ml, Invivogen), LPS (5ng/ml, Sigma), CL075 (1 µg/ml, Invivogen) and CpG (ODN 2216, 7.5 µM, Invivogen) with or without IFN-α (3000 IU/ml, R&D), with or without anti-CD303 and anti-CD304 (Biolegend), with or without 0.1, 1 or 10 µM lenalidomide (Sigma). Cells were cultured for 14 h at 37 ºC, 5% $CO_2$, with addition of Brefaldin A (10 µg/ml, eBioscience) after 3 h. For dead-cell exclusion (usually <30%) cells were stained with Zombie amine dye (Biolegend), surface markers and then intracellular cytokines antibodies after fixation and permeabilization (eBioscience), as above.

**In vitro generation of DCs**. CD34⁺ bone marrow progenitors were purified by fluorescence-activated cell sorting (FACS) (>98% purity) and seeded (3000/well) onto OP9 stromal cells (5000/well) in 96-well U-bottomed plates. Cells were cultured in 200 µl αMEM (Gibco™) supplemented with 1% penicillin/streptomycin (Sigma), 10% fetal calf serum (Gibco), 20 ng/ml granulocyte-macrophage colony-stimulating factor (R&D systems), 100 ng/ml Flt3-ligand (Immunotools), 20 ng/ml stem cell factor (Immunotools), with or without 0.1, 1 or 10 µM lenalidomide (Sigma) or 0.01% dimethyl sulfoxide control. Half the volume of media, with cytokines, was replaced weekly. At day 22, cells were harvested on ice, passed through a 50 µm filter, washed and stained for flow cytometric analysis.

**Statistics**. Graphs were plotted with Prism V5 (GraphPad software Inc.) and mean, 95% confidence interval, s.e.m., s.d., linear regression analysis and Mann–Whitney *U*-tests (two-tailed) were calculated within the software.

**Data availability**. The authors declare that the data supporting the findings of this study are available within the article and its Supplementary Information files, or are available upon reasonable requests to the authors.

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

## Acknowledgements

We gratefully acknowledge the participation of the patients and their families. We thank Rachel Queen from Newcastle University Single Cell Unit for bioinformatics support. We acknowledge the Newcastle University Flow Cytometry Core Facility (FCCF) for assistance with the generation of Flow Cytometry data. This study was funded by Wellcome Trust 101155/Z/13/Z (V.B., U.C.), CGD Society CGDS16/01, EU-FP7 CELL-PID HALTH-2010-261387, EU-FP7 NET4CGD (J.R.), Jeffrey Modell Foundation (S.P., F.H., M.D.), Intramural research program of the NIH Clinical Center (H.S.K., S.D.R.), NIH R21 AI094004 and NIH R03 AI113631 (A.K., K.V.V., H.H.), Research Foundation Flanders (D.B.), Ghent University Hospital Spearhead Initiative for Immunology Research (D.B., M.D., F.H.).

## Author contributions

U.C. conducted experiments, analysed results and wrote the manuscript; A.R. analysed results and wrote the manuscript; T.A. conducted experiments and analysed results; A.G. supervised experiments and appraised the manuscript; D.B., H.S.K., G.J., A.K., K.V.V., S. P., M.D., J.R., H.H. and F.H. provided patient material and appraised the manuscript. S. D.R. conceptualised the study, provided patient material and appraised the manuscript; M.C. conceptualised the study, designed experiments and wrote the manuscript; V.B. conceptualised the study, designed and conducted experiments, analysed results and wrote the manuscript.

## Additional information

**Competing interests:** The authors declare no competing financial interests.

