## [Peer Review File · Nature Communications]

Reviewers' comments:

Reviewer #1 (Transcription factor regulation, DC)(Remarks to the Author):

The study by Cytlak and Bigley describes that similar to mice carrying *Ikzf1* mutations, patients with IKZF1 mutations have abnormal dendritic cell development and function. Twenty patients from 4 families with heterozygous IKZF1 mutations were analyzed and displayed a profound reduction in pDCs, but had an expansion in cDC1. The CD16+ non-classical monocytes were also reduced.

The authors used evidence of the effect of Lenalidomide in Fig. 3 to support the proposed role for the effect of IKZF1 mutation on pDC development, based on patients treated with Lenalidomide in Fig. 3. IKZF1 is reported to be a key target of lenalidomide. The authors showed a negative correlation between lenalidomide dose and IKZF1 protein level, consistent with Lenalidomide reducing IKZF1. However, the authors only showed the correlation between IKZF1 level and DCs/monocytes counts in Figures 3d and e. Functionally, the authors found that peripheral blood monocytes and DCs from patients with IKZF1 mutants were defective in cytokines production in response to a cocktail of TLR agonists. Finally, the authors showed that, in the in vitro culture, lenalidomide treatment lead to reduced IKZF1 level and decreased pDC/expanded cDC1 development.

In summary, the study confirms that IKZF1 is required for proper dendritic cells development in humans, and provides some clues for the pharmacological effect of lenalidomide. However, there are some major flaws in the manuscript that could be improved.

Major points:

1. As members of family F carry a 11-gene deletion, a repeat of the FlowSOM analysis in any of the other three families will better support the the correlation between IKZF1 deficiency and abnormal dendritic cells development.
3. The cocktail of TLR agonists complicates the analysis of dendritic cells function. A separate stimulation with CpG or PolyI:C might have been useful to perform. If this was done, the data could be added

Minor points:

1. As the development of classical monocytes and non-classical monocytes is differentially influenced by IKZF1 deficiency, their function should be separately analyzed in Figure 4a&b.
2. The statement "The most striking overall difference associated with IKZF1 mutation was a relative reduction in pDCs, but increase in cDC1s and cDC2s" from lines 136-138 is not appropriate. The cDC1s are not uniformly increased in Figure 1b.
3. The statement "The expansion of human cDC1s associated with IKZF1 haploinsufficiency was also seen in mice carrying the heterozygous *Ikzf1* L allele, compared with wild-type or homozygous L/L mutants, although this did not reach significance" from lines 324-326 is not appropriate. In reference 22, both the percentage and absolute cell count of CD8 α + DCs are not affected by *Ikzf1* mutation. The difference between cDC1s development in IKZF1 deficient humans and mice should be further discussed in the Discussion Session.
4. The nomenclature of the genes should be consistent in the manuscript. Either IKZF1 or IKAROS should be used.

Reviewer #2 (Myeloma, lenalidomide)(Remarks to the Author):

The authors state that IKAROS/IKZF1, a hematopoietic TF, is absolutely required for human pDC developments with phenotypic and functional analyses. These are very interesting and outstanding findings about the critical role of IKZF1 in DC development and function as for patients with IKZF1 heterozygous mutations, or treated with lenalidomide. The manuscript is well written and the experiments appropriately address the question being asked in the manuscript. There are a few minor comments that should be addressed to further enhance the manuscript.

In consistent with this manuscript, there is preclinical supporting data that lenalidomide synergistically enhances the antitumor effects of DC vaccines in the murine MM model (Lee et al, J Immunother 2015 (38):330-339). In addition to their findings, the enhancement of cDC1 lineage in IKZF1 haploinsufficiency might be linked to accelerate the anti-myeloma effect and adjuvancy of lenalidomide in combination with DC vaccination. You'd better add it in the section of discussion part (line 399-402).

Nowadays, pomalidomide is also one of popular and potent immune modulating drugs (IMiDs) in clinic, I'm just wondering that similar results might be occurred in pomalidomide-treated patients (IKZF1 haploinsufficiency-related DC anomalies after pomalidomide treatment in MM ...).

In spite of several functions of pDCs including anti-viral effect and immune tolerance, pDCs have been also well demonstrated as one of potent anti-tumor DC vaccines through the production of interferon-gamma. Do you think there is any switching factor or threshold to modulate those situations (tolerance or immunity)...or any opinion?

IKZF3 is also key target along with IKZF1, after activation of CRBN-dependent ubiquitination and proteasomal degradation after lenalidomide treatment in MM survival.

Loss of IKZF1 and IKZF3 was required for the therapeutic effect of lenalidomide. Do you think there is any marginal role or possibility of IKZF3 participation in abnormal human DC developments, according to previous reports or your findings?

Reviewers' comments:

We thank the reviewers for their comments and helpful suggestions for improvement. In this revision, we have incorporated the suggestions as outlined point by point below.

Note S2 has been removed as details of Family G are now published in Bogaert et al., 2017, JACI (reference 27; PMID 28927821).

Lines correspond to those in the resubmitted (marked) manuscript.

Reviewer #1 (Transcription factor regulation, DC)(Remarks to the Author):

The study by Cytlak and Bigley describes that similar to mice carrying *Ikzf1* mutations, patients with IKZF1 mutations have abnormal dendritic cell development and function. Twenty patients from 4 families with heterozygous IKZF1 mutations were analyzed and displayed a profound reduction in pDCs, but had an expansion in cDC1. The CD16+ non-classical monocytes were also reduced.

The authors used evidence of the effect of Lenalidomide in Fig. 3 to support the proposed role for the effect of IKZF1 mutation on pDC development, based on patients treated with Lenalidomide in Fig. 3. IKZF1 is reported to be a key target of lenalidomide. The authors showed a negative correlation between lenalidomide dose and IKZF1 protein level, consistent with Lenalidomide reducing IKZF1. However, the authors only showed the correlation between IKZF1 level and DCs/monocytes counts in Figures 3d and e. Functionally, the authors found that peripheral blood monocytes and DCs from patients with IKZF1 mutants were defective in cytokines production in response to a cocktail of TLR agonists. Finally, the authors showed that, in the in vitro culture, lenalidomide treatment lead to reduced IKZF1 level and decreased pDC/expanded cDC1 development.

In summary, the study confirms that IKZF1 is required for proper dendritic cells development in humans, and provides some clues for the pharmacological effect of lenalidomide. However, there are some major flaws in the manuscript that could be improved.

Major points:

1. As members of family F carry a 11-gene deletion, a repeat of the FlowSOM analysis in any of the other three families will better support the the correlation between IKZF1 deficiency and abnormal dendritic cells development.

Thank you for raising this point. We have repeated these analyses on a number of families and find similar results. We agree it would be more stringent to show the data from an individual with a single gene mutation. **Figure 1a and b** now show data from a member of Family B (B5) compared to a travel control (transported with the Family B samples) (**lines 122 and 143; 470-471**).

3. The cocktail of TLR agonists complicates the analysis of dendritic cells function. A separate stimulation with CpG or PolyI:C might have been useful to perform. If this was done, the data could be added.

We agree with the reviewer that this assay is complex. During the workup of this assay, comparisons were performed of single stimulants with the cocktail in healthy control PBMC. This revealed no significant differences in cytokine production for the relevant agonist/cytokine/DC pathways. For example, IL-12 production from cDC1 in response to

Poly I:C alone is equivalent to stimulation with the multi-agonist panel. Similarly, IFN α and TNF production from pDC in response to CpG is comparable (see graphs below; bars represent mean \pm -SEM. P value from paired t-test).

We specifically developed the multi-agonist approach to derive the maximum data from limited clinical material in a reproducible assay. Even with this approach, a minimum total of 6 million PBMC are required to recover sufficient numbers of the rare DC populations (particularly cDC1) for robust analysis in two conditions (no stimulation and all stimulants).

A full description of the assay as a clinical test, including detailed comparison of individual TLR agonists, is being prepared as an independent manuscript. However, we agree with the reviewer that it is an important point to note here and have included additional text in **lines 242-245**.

Minor points:

1. As the development of classical monocytes and non-classical monocytes is differentially influenced by IKZF1 deficiency, their function should be separately analyzed in Figure 4a&b.

Thank you for this suggestion. We have reanalysed the data and were able to identify a non-classical population as CD14-CD11c+ monocytes (CD16 is down-regulated in culture and is not a reliable marker for this subset). The pattern of cytokine production of CD14- non-classical monocytes in the IKZF1 patients was similar to that observed for CD14+ classical monocytes. In keeping with previous literature the magnitude of cytokine production was lower than the CD14+ classical monocytes. These new data are presented in **Figure 4a and b (second row)**.

2. The statement "The most striking overall difference associated with IKZF1 mutation was a relative reduction in pDCs, but increase in cDC1s and cDC2s" from lines 136-138 is not appropriate. The cDC1s are not uniformly increased in Figure 1b.

This text has been modified as suggested, and to describe the new data presented (lines 136-138).

3. The statement “The expansion of human cDC1s associated with IKZF1 haploinsufficiency was also seen in mice carrying the heterozygous *Ikzf1* L allele, compared with wild-type or homozygous L/L mutants, although this did not reach significance” from lines 324-326 is not appropriate. In reference 22, both the percentage and absolute cell count of CD8 α + DCs are not affected by *Ikzf1* mutation. The difference between cDC1s development in IKZF1 deficient humans and mice should be further discussed in the Discussion Session.

This text has been modified as suggested (lines 335-337)

4. The nomenclature of the genes should be consistent in the manuscript. Either IKZF1 or IKAROS should be used.

This has been corrected.

Reviewer #2 (Myeloma, lenalidomide)(Remarks to the Author):

The authors state that IKAROS/IKZF1, a hematopoietic TF, is absolutely required for human pDC developments with phenotypic and functional analyses. These are very interesting and outstanding findings about the critical role of IKZF1 in DC development and function as for patients with IKZF1 heterozygous mutations, or treated with lenalidomide. The manuscript is well written and the experiments appropriately address the question being asked in the manuscript. There are a few minor comments that should be addressed to further enhance the manuscript.

In consistent with this manuscript, there is preclinical supporting data that lenalidomide synergistically enhances the antitumor effects of DC vaccines in the murine MM model (Lee et al, J Immunother 2015 (38):330-339). In addition to their findings, the enhancement of cDC1 lineage in IKZF1 haploinsufficiency might be linked to accelerate the anti-myeloma effect and adjuvancy of lenalidomide in combination with DC vaccination. You'd better add it in the section of discussion part (line 399-402).

Thank you for drawing our attention to this interesting study which is now included in the discussion (lines 417-420, Ref 48).

Nowadays, pomalidomide is also one of popular and potent immune modulating drugs (IMiDs) in clinic, I'm just wondering that similar results might be occurred in pomalidomide-treated patients (IKZF1 haploinsufficiency-related DC anomalies after pomalidomide treatment in MM ...).

This is an interesting suggestion. Although not included in the original manuscript, we had recruited 2 patients treated with pomalidomide to the study. These data are now added to **Figure 3d**, and patient details in **table S3**. As suggested by the reviewer, these data are consistent with pomalidomide-induced IKZF1 deficiency and lower pDC numbers (lines 202, 231-232 and 235).

In spite of several functions of pDCs including anti-viral effect and immune tolerance, pDCs have been also well demonstrated as one of potent anti-tumor DC vaccines through the production of interferon-gamma. Do you think there is any switching factor or threshold to modulate those situations (tolerance or immunity)...or any opinion?

The reviewer raises the interesting issue that pDC function (immunity versus tolerance) is likely to be context dependent. It may be possible to isolate the immunogenic functions of pDCs for therapeutic benefit in myeloma but we are not aware of any conditions in which this has been demonstrated. This would certainly be worthy of exploration but we feel that a discussion of this is beyond the scope of our study.

IKZF3 is also key target along with IKZF1, after activation of CRBN-dependent ubiquitination and proteasomal degradation after lenalidomide treatment in MM survival. Loss of IKZF1 and IKZF3 was required for the therapeutic effect of lenalidomide. Do you think there is any marginal role or possibility of IKZF3 participation in abnormal human DC developments, according to previous reports or your findings?

This is point is well taken. Our data are unable to exclude an IKZF3-dependent effect of ImiDs on DCs. The discussion is modified to include this statement (**lines 371-372**). However, in support of a predominantly IKZF1 mechanism we note the following:

- 1) There is a linear relationship between IKZF1 protein level and pDC number
- 2) A similar effect is seen in IKZF1 deficiency whether due to heterozygous mutation or drug therapy.
- 3) IKZF1 is expressed at more than 100 times the level of IKZF3 in pDC (and at a similarly higher level in other DC subsets) – these data are now shown in **Figure 1c** and **lines 156/7 and 212-214**)
- 3) No role for IKZF3 has been described in murine models of DC development. These points are also discussed (**lines 373-374**). We also note that cereblon-dependent suppression of IRF4 has been observed and include this in the discussion (**lines 369-371**).

REVIEWERS' COMMENTS:

Reviewer #1 (Remarks to the Author):

The revised study by Cytlak and Bigley has addressed the comments I had. It is a bit disappointing that the authors chose to resort to a "data not shown" for the more major I had about the "kitchen sink" activation approach used, and the authors have avoided doing what I suggested. It may be that they think this is OK, since they didn't see differences in WT responses when they did single modes of stimulation. I guess I don't care, since this study is essentially a confirmation of a known observation. But it's the type of behavior that can prevent the discovery of novel findings, if for example, the factor impairs one pathway in the KO, but not another pathway. As confirmation, this study is fine.

Reviewer #2 (Remarks to the Author):

The manuscript is really well written and the authors have good responses to us.

** See Nature Research's author and referees' website at www.nature.com/authors for information about policies, services and author benefits

Reviewers' comments:

Reviewer #1 (Remarks to the Author):

The revised study by Cytlak and Bigley has addressed the comments I had. It is a bit disappointing that the authors chose to resort to a "data not shown" for the more major I had about the "kitchen sink" activation approach used, and the authors have avoided doing what I suggested. It may be that they think this is OK, since they didn't see differences in WT responses when they did single modes of stimulation. I guess I don't care, since this study is essentially a confirmation of a known observation. But it's the type of behavior that can prevent the discovery of novel findings, if for example, the factor impairs one pathway in the KO, but not another pathway. As confirmation, this study is fine.

As previously noted, there was insufficient primary patient material available to perform single stimulation assays. However, we have now included data showing cytokine elaboration in response to single TLR agonists in healthy controls. These results are shown in supplementary data. We share the reviewer's frustration that experimental options may be limited when working with primary material. The experiments we devised were carefully planned to extract the maximum relevant information but we recognise that this approach is not exhaustive.

Reviewer #2 (Remarks to the Author):

The manuscript is really well written and the authors have good responses to us.